# Psychosocial Predictors of Job Satisfaction in Nursing: Insights from a Spanish Hospital Setting

**DOI:** 10.3390/bs15030274

**Published:** 2025-02-26

**Authors:** Alejandra Trillo, Alberto Ortega-Maldonado, Beatriz Lopez-Pena, Francisco D. Bretones

**Affiliations:** 1Department of Social Psychology, Faculty of Labour Relations and Human Resources, Universidad de Granada, 18012 Granada, Spain; atrillo@ugr.es; 2Department of Psychology, Faculty of Business and Communication, Universidad Internacional de La Rioja, 26006 Logroño, Spain; alberto.ortega@unir.net; 3Research Group Sustainable Talent Development, Faculty Business, Finance & Marketing, The Hague University of Applied Sciences, 2521 EN Den Haag, The Netherlands; b.lopezpena@hhs.nl

**Keywords:** job satisfaction, psychological empowerment, affective commitment, emotional exhaustion, nursing

## Abstract

Nurses play a key role in healthcare systems, yet their job satisfaction is often challenged by factors such as emotional exhaustion and organisational dynamics. This study examines the relationship between psychological empowerment, affective commitment, and emotional exhaustion as predictors of job satisfaction in a sample of 150 Spanish nurses. Data were collected using validated questionnaires measuring these constructs, and mediation analyses were conducted using the PROCESS macro. Results indicated that psychological empowerment positively influenced job satisfaction both directly and indirectly through affective commitment. However, emotional exhaustion did not significantly mediate this relationship, suggesting that contextual factors such as workload may override its effects. This research contributes to the understanding of job satisfaction among healthcare professionals and highlights the importance of empowerment and affective engagement. Practical implications for hospital management include fostering psychological empowerment through organisational strategies that promote autonomy, competence, and meaningful involvement, which could improve nurses’ well-being and organisational performance.

## 1. Introduction

Job satisfaction, understood as the balance between job expectations and the rewards offered by the organisation, directly influences service quality, customer satisfaction, organisational development, and has a significant impact on subjective well-being and mental health ([35]; [40]). Understanding the factors that contribute to nurses’ satisfaction is particularly relevant as the public health crisis resulting from the COVID-19 pandemic intensified the emotional and organisational demands on nurses and had a significant impact on their well-being and commitment ([7]). Thus, the public health emergency highlighted the need to re-examine traditional theoretical models to adapt them to the new challenges faced by nurses in a post-pandemic context.

Over the years, numerous studies have analysed the relationship between different psychosocial factors and job satisfaction in nursing ([33]). Among these, psychological empowerment has been identified as a key predictor, as it gives workers greater autonomy and control over their functions, improves their perception of self-efficacy, and reduces job dissatisfaction ([25]; [41]). However, recent studies have raised questions about the direct relationship between psychological empowerment and emotional exhaustion, as some findings suggest that organisational factors, such as workload, may moderate the impact of psychological empowerment on reducing emotional exhaustion ([11]).

This is particularly relevant in nursing, a profession characterised by high emotional demands and chronic work stress. In fact, nurses have a 54% prevalence of burnout worldwide ([54]), which highlights the importance of investigating how psychosocial variables may influence their job satisfaction and well-being.

Therefore, this study aims to complement and update the current literature, as well as to extend and update knowledge on job satisfaction in nursing, by jointly analysing psychological empowerment, affective commitment, and emotional exhaustion in a context where working conditions have changed significantly in recent years due to the global health crisis. This approach will not only allow us to contrast previous findings in a specific socio-cultural reality but will also help to clarify discrepancies in the literature on the role of emotional exhaustion as a mediator.

## 2. Theoretical Background and Hypothesis Development

The study of psychological empowerment has gained relevance, especially after the turning point that the COVID-19 health emergency brought about for health organisations ([27]).

Psychological empowerment can be defined as the set of internal perceptions and beliefs that enable employees to influence their work environment and work outcomes ([16]). This concept includes dimensions such as job meaning (the fit between job requirements and beliefs or the value of a job goal compared to an individual’s own ideals or standards), competence (an individual’s confidence or belief in his or her ability to perform activities competently), autonomy (the sense of choice or control over one’s work), and impact (the sense of being able to influence important work outcomes) ([47]). For nurses, psychological empowerment implies a strong sense of autonomy and efficacy in caring for patients, enabling them to take initiative and respond creatively to the challenges they face in their daily work ([43]).

Psychological empowerment is closely related to job satisfaction, which is understood as an individual’s attitude towards their work ([39]). Specifically, some authors ([10]) argue that when nurses perceive that they have control over their tasks and the ability to influence decisions related to their work, they tend to experience higher levels of job satisfaction. Thus, psychological empowerment acts as an internal motivator that fosters positive beliefs in nurses about their ability to make a meaningful contribution to the work environment. These positive beliefs, in turn, increase job satisfaction ([26]).

Based on several studies that have found a significant correlation between the two variables in nurses ([52]; [34]), we hypothesise the following:

**H1.** *Psychological empowerment (PE) predicts job satisfaction (JS)*.

### 2.1. The Mediating Effect of Affective Commitment on the Relationship Between PE and JS

Another variable that has been linked to job satisfaction is organisational commitment or the worker’s psychological attachment to his/her organisation ([2]). Affective commitment, which is considered stronger and more consistently linked to desirable organisational outcomes ([18]), is defined as the emotional bond between the employee and the company and is considered the most consistent form of commitment within the broader framework of organisational behaviour. Affective commitment can be expected to yield stronger, more positive organisational outcomes because employees who are effectively committed to an organisation choose to remain its employees, rather than feeling obliged to (normative commitment) or staying because leaving would be costly (continuance commitment). The COVID-19 pandemic significantly influenced affective commitment in the nursing profession by changing the perceptions of organisational support, job security, and professional identity ([14]).

However, previous research has highlighted the role of psychological empowerment in promoting affective commitment. In particular, some authors ([22]) have shown that active participation at work, facilitated by empowering practices, can significantly contribute to the development of affective commitment. Thus, when organisations empower their employees, allowing them to demonstrate their competence and visualise themselves achieving high levels of performance, employees tend to show higher levels of commitment and greater motivation to perform ([38]; [6]).

Similarly, numerous studies have established a link between affective commitment and employee satisfaction. For example, some studies ([42]) found that employees with strong affective commitment, who perceive their relationship with the organisation as equitable and satisfying, tend to reciprocate by developing positive work attitudes, such as job satisfaction. Similarly, others ([3]) argue that highly affectively committed employees are more willing to put extra effort into their tasks, which increases their overall satisfaction. Thus, employees with strong affective ties to their organisations experience higher satisfaction, as their emotional attachment fosters positive perceptions of their work environment. Based on this evidence, we formulate the following hypothesis:

**H2.** *Affective commitment mediates the relationship between psychological empowerment and job satisfaction*.

### 2.2. The Mediating Effect of Emotional Exhaustion on the Relationship Between PE and JS

However, the relationship between PE and JS may be influenced by other variables, such as emotional exhaustion. As mentioned above, healthcare professionals are particularly prone to developing emotional exhaustion, which is one of the dimensions of burnout ([9]; [53]), especially in the aftermath of the pandemic ([44]). Authors ([29]) define burnout as a prolonged response to chronic emotional and interpersonal stressors on the job. Among the different signs of this syndrome, emotional tiredness or exhaustion represents the basic dimension ([36]; [50]), which refers to the feeling of lacking emotional and physical resources to face the perceived stressors ([21]).

Thus, nurses largely view their work environment as stressful, which can lead to emotional exhaustion and decreased job satisfaction ([1]). However, it has been found that psychological empowerment can be used as an organisational initiative that can reduce feelings of powerlessness ([45]) and therefore also reduce levels of job dissatisfaction. This is because nurses have to make very quick decisions to respond to patients’ needs, so if they are included in the decision-making process, they will decrease stress related to formal barriers and therefore receive more job satisfaction ([24]).

Therefore, based on previous studies ([13]; [12]), which demonstrated the mediating role of emotional exhaustion in the relationship between job satisfaction and psychological empowerment, we hypothesise the following:

**H3.** *Emotional exhaustion (EE) mediates the relationship between psychological empowerment (PE) and job satisfaction (JS)*.

Referring to the relationship between the variables, we propose the following model, as shown in Figure 1.

## 3. Research Methods

### 3.1. Sample and Data Collection

Given the impossibility of precisely determining the total size of the target population, the minimum sample size was calculated using the formula proposed by [48] ([48]): *n* = 50 + 8 × *m* (where ‘*m*’ is the number of independent variables included in the study); the minimum required sample size was 74. To prepare the sample, the questionnaire was distributed to all nursing staff at one University Hospital located in southern Spain. The inclusion criteria were currently working as a nurse and to have at least 1 year of professional experience at the time of answering the survey. A total of 280 questionnaires were sent out, with 150 valid responses (53.57%). Therefore, this study had a larger sample size than the minimum suggested by the established parameters.

Their ages ranged from 24 to 62 (mean age of 46), and they had been working at the hospital for an average of 17 years. The sample was 78.8% female and 21.2% male. Finally, 61.4% of the sample was regular staff with permanent contracts; 29.5% were temporary employees, and 9.1% had another administrative status. Participation in the study was voluntary. Throughout all the research phases (data collection, analysis, and interpretation), the subjects remained anonymous, and their personal data were protected.

### 3.2. Measures

As for the measuring instruments, we used the following standardised questionnaires.

#### 3.2.1. Job Satisfaction

Job satisfaction was measured using the short version of the Spanish 20-item Minnesota Satisfaction Questionnaire (MSQ) ([51]), measured on a 1–7 scale, where 1 means ‘very dissatisfied’ and 7 means ‘very satisfied’. Item responses are summed or averaged to create a total score; the lower the score, the lower the level of job satisfaction. The MSQ survey has been extensively utilised in several studies with reliability coefficient ratings that ranged from 0.78 to 0.93 ([15]). The Cronbach alpha reliability index in our study was 0.89. An example item from the MSQ includes ‘The opportunity this job gives me to be “somebody” in society’.

#### 3.2.2. Affective Commitment (AC)

Affective commitment was assessed using the Spanish adaptation ([4]) of the Affective Commitment Scale (ACS) ([30]). This version has six items; an example includes “I am proud to tell others I work at my organization”. The responses are Likert-type answers of 7 points from ‘Completely disagree’ (1) to ‘Strongly agree’ (7). This scale has been used in previous studies and has shown good psychometric properties ([18]). In the present study, the internal consistency (Cronbach’s alpha) was 0.72.

#### 3.2.3. Psychological Empowerment

Psychological empowerment was evaluated with the Spanish adaptation of the Psychological Empowerment Inventory (PEI) ([47]) form with 11 items ([5]), with one being “I have confidence in my ability to do my job”. The responses were obtained on a Likert-type scale, where 1 means ‘little’ and 7 ‘much’. The higher the score, the more nurses perceived themselves as being empowered by their organisation. For this study, the Cronbach alpha of the total scale was 0.87.

#### 3.2.4. Emotional Exhaustion

Lastly, the variable of emotional exhaustion was assessed using the Maslach Burnout Inventory (MBI) ([28]) in its Spanish version ([17]). This five-item scale measures how often one feels emotionally overextended and exhausted by one’s work. A sample item being “I feel emotionally drained in my job”. The inventory uses a 7-point scale from 0 (never) to 6 (always). In the present study, the internal consistency (Cronbach’s alpha) was 0.82. This questionnaire is one of the most widely used in studies of this syndrome ([32]).

## 4. Data Analysis

To test each hypothesis, we conducted various statistical analyses with SPSS 25.0 and PROCESS v3.4. Initially, common method bias was assessed using an exploratory factor analysis, followed by determining the mean and standard deviation of each variable and performing a Pearson correlation analysis. Finally, we adopted the approach of [37] ([37]) using PROCESS macromodel 6 in order to request 5000 bootstrap resamples of the obtained data. We also derived 95% bias-corrected confidence intervals (CI). This software, which is widely used in the behavioural sciences, was chosen for its ability to assess multiple mediating pathways simultaneously and to conduct mediation and moderation analyses with bootstrap-based estimates, which increases the precision of confidence intervals and reduces the likelihood of Type I error ([37]). Furthermore, compared to structural equation models (SEMs), PROCESS is better at focusing on specific mediating factors rather than general latent construct relationships ([19]). Finally, model 6 was selected for its ability to test for sequential mediation effects, with affective commitment and emotional exhaustion creating an indirect causal chain in relation to psychological empowerment and job satisfaction among healthcare workers.

## 5. Results

### 5.1. Common Method Bias Test

An exploratory factor analysis was performed to assess the presence of a possible common method bias ([49]). The results indicated that 12 factors had eigenroot values greater than one. In addition, the first common factor explained only 20.81% of the cumulative variance, below the commonly accepted threshold of 40.00%. These results indicate that the data comprising this study are not affected by common method bias.

### 5.2. Examining Descriptive Statistics

Pearson correlation analysis was used to examine the relationships between psychological empowerment, job satisfaction, affective commitment, and emotional exhaustion. A detailed summary of the descriptive statistics for each variable, including the mean (M), the standard deviation (SD), and correlation coefficients, is presented in Table 1.

The analysis revealed that psychological empowerment (PE) was significantly positively correlated with job satisfaction (JS) (r = 0.365, *p* < 0.01) and affective commitment (AC) (r = 0.281, *p* < 0.01), suggesting that higher levels of psychological empowerment are associated with higher levels of job satisfaction and affective commitment. In contrast, psychological empowerment showed a negative association with emotional exhaustion (EE), although the correlation did not reach statistical significance (r = −0.720).

Furthermore, job satisfaction (JS) showed a significant positive correlation with affective commitment (AC) (r = 0.549, *p* < 0.01), indicating that employees with higher job satisfaction tend to show stronger affective commitment to their organisation. Furthermore, job satisfaction was significantly negatively correlated with emotional exhaustion (AL) (r = −0.329, *p* < 0.01), highlighting that lower levels of emotional exhaustion are linked to higher satisfaction.

Finally, affective commitment (AC) was significantly negatively correlated with emotional exhaustion (r = −0.355, *p* < 0.01), suggesting that higher affective commitment is associated with lower emotional exhaustion.

Regarding the other variables in Table 1, there was no significant correlation with the sociodemographic (gender and age) or organisational (job seniority) variables. A comparative analysis of mean values with Student’s *t*-test reflected a significant association between burnout and gender, since male subjects had higher levels of burnout than female subjects (*t*(130) = 2.27; *p* < 0.05).

### 5.3. Testing the Serial Multiple Mediation Model

After verifying the reliability and validity of the instruments, as well as the intercorrelation between each variable, a structural model analysis was carried out. We have proved this hypothesis using bootstrapped mediation tests recommended by [37] ([37]). This test has been recommended as one of the most accurate tests of mediation, especially when sample sizes are small ([8]). In addition, this test does not perform an ordinal sampling distribution of indirect effects ([20]) and displays the path coefficient (β) to determine the contribution of each predictor variable to the endogenous variable. In addition, R-squared values were used to assess the explanatory power of the model. The results of the hypothesis tests are presented in Table 2 and Figure 2.

Regarding the direct effect, Table 2 shows that the variable psychological empowerment (PE) presents a strong and statistically significant direct effect on job satisfaction (JS), which supports our first hypothesis.

Additionally, hypotheses 2 and 3 suggested that affective commitment and emotional exhaustion will measure the relation between psychological empowerment and job satisfaction. As shown in Figure 2, the mediation tests following the [37] standards provide partial support. The 95% bootstrap CIs do not contain zero, confirming the presence of statistically significant indirect effects of affective commitment but not those of emotional exhaustion.

In order to test the significance of the mediating effect of AC, the Sobel test was performed. The test result (z = 3.19, *p* = 0.001) shows the AC effect as a mediating variable by significantly reducing the indirect effect of PE on JS (*a* = 0.268 and *b* = 0.408).

## 6. Discussion

Our study allowed us to gain a more comprehensive view of the importance of psychological empowerment for job satisfaction and how this relationship is influenced by affective commitment and emotional exhaustion in a sample of Spanish nurses.

First, the results of our study provide evidence to support hypothesis one, which postulates that psychological empowerment (PE) predicts job satisfaction (JS). Consistent with previous research ([10]; [26]; [52]), it has been observed that when workers perceive greater autonomy and control over their work, their job satisfaction tends to increase. However, this finding is particularly relevant in the post-pandemic context, as changes in hospital management have affected health professionals’ perceptions of control ([46]). One possible explanation for the strength of this relationship is that psychological empowerment may partially counteract the negative effects of work-related stress by allowing nurses to cope with the challenges of their environment. However, the organisational context still plays a key role, as a highly demanding work environment, such as during and after COVID-19 ([31]), may reduce the benefits of psychological empowerment.

Another conclusion that can be derived from this study is that affective commitment plays a significant role as a mediator between psychological empowerment and job satisfaction. This is important because it highlights the effect on job satisfaction, so companies that want to implement empowerment policies with their employees should also consider the affective aspects of healthcare workers to ensure a greater impact on their job satisfaction. It was also found that affective commitment preceded job satisfaction and was thus a good predictor of it, which confirms the results of other studies ([42]) that used different sample populations.

However, one of the most surprising findings of our study was the lack of association between psychological empowerment and emotional exhaustion. This finding contrasts with previous studies that have found an inverse relationship between these two variables in nurses ([54]; [12]).

This lack of mediation of emotional exhaustion could be explained by the conservation of resources theory ([23]), which suggests that prolonged job stress may reduce the ability of employees to benefit from internal resources such as psychological empowerment. Thus, in a post-pandemic context where nurses’ workloads have increased significantly without a parallel increase in organisational resources, psychological empowerment may not have been sufficient to reduce emotional exhaustion. This finding suggests that hospitals should not only promote psychological empowerment but also provide adequate working conditions for its effects to be sustained.

## 7. Conclusions

Our findings reinforce the importance of psychological empowerment as a predictor of job satisfaction but suggest that its effect is highly dependent on affective commitment. However, the mediation of emotional exhaustion was not significant, suggesting that its impact on job satisfaction could be conditioned by other contextual factors, such as workload or the availability of organisational resources.

From a theoretical point of view, this study provides additional evidence on the relationship between psychological empowerment and job satisfaction in a specific cultural and organisational context, which allows us to contrast results obtained in other countries and enrich our knowledge on the psychosocial mechanisms that influence the well-being at work of healthcare professionals. Furthermore, the research extends the literature by analysing the simultaneous interaction of psychological empowerment, affective engagement, and emotional exhaustion in a mediational model, which has not been sufficiently explored in previous studies.

In practical terms, the results suggest that healthcare institutions can improve the job satisfaction of their nurses through strategies that foster psychological empowerment and affective engagement. This implies the implementation of programmes that promote autonomy, such as participation in decision-making and the recognition of individual effort. It is also recommended to design specific interventions to strengthen affective commitment, such as improving working conditions and creating organisational environments that favour identification with the institution. Since emotional exhaustion was not found to be a significant mediator, future organisational strategies should focus on identifying other factors that could moderate its impact, such as social support, workload, or flexibility in the working day.

## 8. Limitations and Future Recommendations

While the findings of our study are highly valuable and promising, we acknowledge certain limitations that should be considered in future research.

One of the main limitations lies in the cross-sectional nature of the data collected, which prevents us from establishing definitive causal relationships between the variables analysed. Therefore, future research should address this limitation by designing longitudinal studies, which would allow for a more precise exploration of the temporal and dynamic relationships between these variables.

Another important limitation concerns the sample size and its representativeness. The data were collected exclusively from nurses working in a single hospital, which limits the generalisability of the results to other populations and contexts, and future studies should extend the sample to different hospitals, regions, and sectors of the healthcare system in order to validate and contrast the findings in a wider range of work settings.

In addition, traditional research approaches often examine factors in isolation or using linear mediation models (e.g., PE → AC → JS). However, relationships among PE, AC, EE, and JS are likely nonlinear. Future research could benefit from studying the variables as a holistic system, using a system dynamics approach. Analysing the relationships between the variables would allow for a deeper exploration of how they dynamically interact over time and how their interdependencies influence overall job satisfaction.

It would also be advisable for future research to consider the inclusion of other contextual or personal variables that may play a mediating or moderating role in the observed relationships. For example, factors such as social support, organisational culture, or individual characteristics such as resilience or self-efficacy could enrich our understanding of the mechanisms underlying the relationships between empowerment, engagement, burnout, and job satisfaction. Furthermore, it would be particularly interesting to analyse whether contractual conditions, such as instability and job insecurity, influence these relationships, especially since almost 40% of our sample is made up of staff with unstable employment.

Finally, given the specific focus on the hospital setting, it would also be relevant to investigate whether these findings are replicated in other occupational settings, with a particular focus on those that share characteristics of high emotional demand, such as education or emergency services. This approach would broaden the applicability of the findings and strengthen the practical implications for the design of interventions aimed at improving workers’ well-being.

## Figures and Tables

**Figure 1 behavsci-15-00274-f001:**
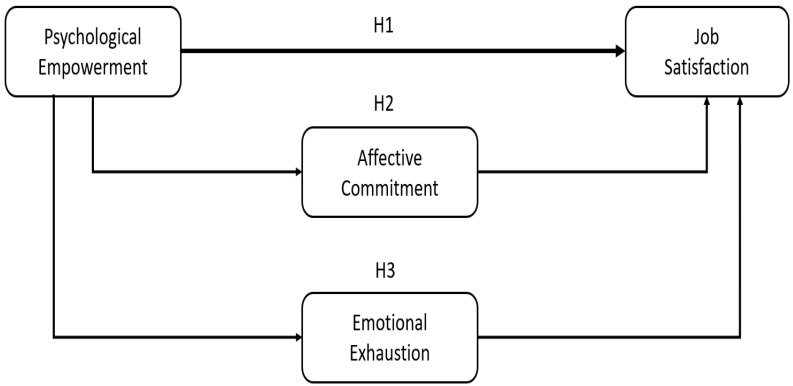
Hypothesised serial multiple mediation model. Source(s): author’s own work.

**Figure 2 behavsci-15-00274-f002:**
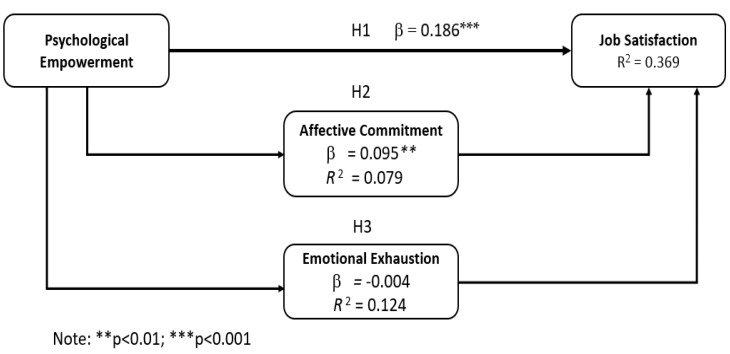
Multiple mediation model. Source(s): author’s own work.

**Table 1 behavsci-15-00274-t001:** Means, standard deviations, and correlations of variables in the model.

	M	SD	PE	JS	AC	EE	JSE	GE
Psychological Empowerment (PE)	4.99	1.165						
Job Satisfaction (JS)	4.86	0.787	0.365 **					
Affective Commitment (AC)	4.30	0.677	0.281 **	0.549 **				
Emotional Exhaustion (EE)	3.84	0.949	−0.720	−0.329 **	−0.355 **			
Job Seniority (JSE)	16.07	10.16	0.031	−0.011	0.018	0.064		
Gender (GE)	0.79	0.406	0.068	0.098	0.000	−0.132	0.120	
Age (AG)	44.57	9.45	−0.029	−0.063	0.063	0.137	0.697 **	0.087

Note: ** *p* < 0.01.

**Table 2 behavsci-15-00274-t002:** Validation of the research hypothesis.

Hypothesis	β	CI	*p*	T Statistics	Sig
Direct effects					
H1: PE-> JS	0.186	(0.778; 0.2938)	0.000	3.400	Yes
Indirect effects					
H2: PE->AC->JS	0.095	(0.036; 0.168)			Yes
PE->AC	0.268	(0.119; 0.417)	0.001	3.551	Yes
AC->JS	0.356	(0.235; 0.476)	0.000	5.812	Yes
H3: PE->EE ->JS	−0.004	(−0.028; 0.021)			No
PE->EE	0.043	(−0.190; 0.277)	0.7146	0.367	No
EE->JS	−0.085	(−0.160; −0.009)	0.028	−2.223	Yes

Note: PE = psychological empowerment; JS = job satisfaction; AC = affective commitment; EE = emotional exhaustion.

## Data Availability

The datasets generated during and/or analysed during the current study are available from the corresponding author on reasonable request.

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
