# Peer review of "Psychosocial Predictors of Job Satisfaction in Nursing: Insights from a Spanish Hospital Setting"

_behavsci, 2025, doi:10.3390/bs15030274_

Round 1
Reviewer 1 Report
Comments and Suggestions for Authors
Please refer to the comments in the PDF review.
Hypothesis 3 should possibly be changed - it is not clear theoretically and it is not explained well.

There are grammatical and spelling errors.
Reviewer 2 Report
Comments and Suggestions for Authors
Thank you for the opportunity to review the paper Psychosocial Predictors of Job Satisfaction in Nursing: Insights from a Spanish Hospital Setting
Here are my comments and main concerns:
1. Since the authors themselves state multiple times in the introductory and theoretical parts that a large number of research have already looked at the relationships between the variables being studied and work satisfaction, the paper lacks originality. This brings up the essential query: what is the goal of this research? To set it apart from earlier studies, the novel features need to be emphasized. The paper runs the risk of being interpreted as a simple replication without this. I suggest changing the introduction to highlight this study's distinctive contributions.
2. The theoretical framework and hypothesis development sections seem to be a collection of previous studies, followed by an intention to retest the same relationships. This lacks a clear justification for why these relationships are being re-examined. Please provide a stronger theoretical rationale and clarify the added value of retesting these associations.
3. As acknowledged by the authors, another main limitation of the study is its cross-sectional design. Mediation analyses are more suitable for longitudinal studies, as they imply causal/temporal relationships between variables. Using this method with a cross-sectional dataset undermines the robustness of the findings. Furthermore, analyzing the data using more robust methodologies, such as structural equation modeling (SEM), would have been more appropriate, which could provide a stronger foundation for interpreting results.
4. A more critical assessment of the findings that links them to pertinent literature is needed in the discussion section. Without providing further insights or interpretations, the talk seems to rehash the findings. The study would be strengthened by a more interpretative and critical approach.
5. The paper lacks a section dedicated to practical and research implications. How can professionals and researchers use the findings to improve nurses’ working conditions and job satisfaction? The introduction states that the study "provides empirical evidence to inform intervention strategies," but this aspect is entirely missing from the paper. Please provide further details about how the results can be used in future studies and real-world applications. Those are links to my comment. 1. about the significance of comprehending the purpose of this study in contrast to what is previously established in the literature.
6. The paper is missing a concluding paragraph.
In conclusion, the main limitation of this study is that it seems to be a mere repetition of previous studies and, therefore, is not innovative. The main conclusions, as well as the practical implications for research and practice, are not clear. The analyses performed are not very robust and only correlational, and the results in the discussions were not analyzed critically enough. I hope you can improve your work by working on these aspects.
bests,
Round 2
Reviewer 1 Report
Comments and Suggestions for Authors
none
Reviewer 2 Report
Comments and Suggestions for Authors
I thank the authors for revising the paper following my comments. In this version, the manuscript can be accepted for publication